# UV-Vis Sintering Process for Fabrication of Conductive Coatings Based on Ni-Ag Core–Shell Nanoparticles

**DOI:** 10.3390/ma16227218

**Published:** 2023-11-17

**Authors:** Anna Pajor-Świerzy, Lilianna Szyk-Warszyńska, Dorota Duraczyńska, Krzysztof Szczepanowicz

**Affiliations:** Jerzy Haber Institute of Catalysis and Surface Chemistry Polish Academy of Sciences, Niezapominajek 8, 30-239 Kraków, Poland; ncszyk@cyf-kr.edu.pl (L.S.-W.); dorota.duraczynska@ikifp.edu.pl (D.D.); krzysztof.szczepanowicz@ikifp.edu.pl (K.S.)

**Keywords:** nickel–silver core–shell nanoparticles, UV-Vis sintering, conductive coatings

## Abstract

The UV-Vis sintering process was applied for the fabrication of conductive coatings composed of low-cost nickel–silver (Ni@Ag) nanoparticles (NPs) with core–shell structures. The metallic films were formed on a plastic substrate (polyethylene napthalate, PEN), which required their sintering at low temperatures to prevent the heat-sensitive polymer from destroying them. The UV-Vis sintering method, as a non-invasive method, allowed us to obtain metallic coatings with good conductivity at room temperature. In optimal sintering conditions, i.e., irradiation with a wavelength of 350–400 nm and time of 90 min, conductivity corresponding to about 30% of that of bulk nickel was obtained for the coatings based on Ni@Ag NPs.

## 1. Introduction

The development of the global electronics industry market is associated with an increase in demand for new materials characterized by high conductivity, reproducibility, and low production costs. In this context, the low-temperature and efficient technologies used in the process of fabricating printed electronic circuits based on low-cost metallic nanoparticles have attracted significant interest. This issue is important not only from an economic point of view but also with regard to the fabrication of flexible electronic devices, such as organic light-emitting diode (OLED), radio frequency identification (RFID) tags, flexible solar cells, displays, and wearable electronics [1,2,3,4,5], in which, due to the use of heat-sensitive substrates, a low sintering temperature is required. The application of common polymers, such as polyethylene terephthalate (PET), polyimide (PI), and polyethylene terephthalate glycol (PETG), as substrates is a pathway to a wide variety of efficient and cheap electronic devices.

The most important component of inks used for the fabrication of printed electronic circuits and devices is an active material, which is responsible for the conductive properties of printed patterns. Metallic nanoparticles (NPs) can lower the temperature of the preparation process of conductive materials due to their much lower melting point compared to their form at macro size. Therefore, they have commonly been used as the main compound of inks for the printing process of electronic patterns [1,4,6,7,8,9]. The ink based on metallic NPs is deposited onto proper substrates using a printing or coating process, and as a result of this process, a pattern composed of conducting metallic NPs capped with insulating organic stabilizers is formed. Due to the presence of such insulating organic materials, the number of percolation paths is limited, and the resistivity of the printed pattern is usually too high for use in practical applications. This obstacle is conventionally overcome via a post-printing sintering process, which is mainly achieved by heating the printed substrates to temperatures usually higher than 200 °C [9,10,11]. This sintering phenomenon is usually attributed to the reduced melting point of NPs and the high self-diffusion coefficient of their atoms [12]. However, due to the sensitivity of flexible substrates (papers and many plastics) to high temperatures, treatment involving heating at high temperatures is not required [1]. Therefore, there is a great need for a technology that will enable the sintering of the metallic NPs at low temperatures or even without heating to avoid the destruction of the heat-sensitive substrates, as well as lower the costs of the process of fabricating printed electronics.

As promising alternatives to thermal sintering, chemical [13,14,15,16] and radiation-induced sintering have been considered [17,18,19]; accordingly, the chance of thermal damage can be significantly minimalized. Among radiation-induced sintering methods, laser and ultraviolet (UV/UV-Vis) sintering processes were proposed. The wavelength of a laser can be adjusted to specific particle/substrate system; moreover, the pulse width of a laser can be as short as a femtosecond. The laser-sintered thin films composed of Cu NPs with good electrical properties were obtained by Known et al. [17]. They noticed that such a method ensures the rapid fabrication of conductive films with the suppression of the oxidation process of Cu NPs. Noha et al. [18] compared nanosecond to femtosecond laser sintering of Ag NPs films deposited on polyethylene terephthalate (PET) substrates, putting emphasis on the sintering mechanisms and the properties (electrical conductivity, flexibility, and adhesion strength) of the sintered film. The lower resistivity of laser-sintered Ag film was obtained after femtosecond laser sintering in comparison to the nanosecond process. The lowest resistivity of the Ag film sintered with the femtosecond laser sintering process was 7.07 μΩ·cm. The parameters of the laser sintering process of circuits based on Cu NPs on PET substrates were optimized by Hernandez-Castaneda et al. [19]. The lowest values of resistivity were obtained for the green laser (λ = 532 nm) at the lowest laser fluence and the diode laser (λ = 808 nm) with the lowest power density. This laser fabrication process was also applied for the preparation of Ni electrodes combined with a colorless polyimide substrate, which offers significant potential for wider applications using high-temperature transparent heaters [20]. Bischoff et al. [21] produced the copper layers from the ball milling of a CuO NP precursor with a sheet resistivity of 0.2780 Ω/sq via the femtosecond laser reductive sintering process. Besides its application in the electronics industry, the laser sintering process can be also used for the fabrication of biosensors. The micropatterns based on Cu and Ni on the surface of glass-ceramics via the reduction of CuO and NiO NPs using the selective laser sintering (SLS) method were manufactured by Tumkin et al. [22]. The fabricated electrode with good selectivity, long-term stability, and reproducibility can be used as the potential non-enzymatic glucose sensor.

Even if laser sintering is the fastest and most effective method for obtaining conductive structures, it has a small local sintering area and requires an expensive and sophisticated system. To solve this problem, the UV/UV-Vis sintering method of films composed of metallic nanoparticles was also proposed. After one UV/UV-Vis sintering run, a resistivity of 8.8 μΩ·cm, similar to that obtained after the thermal sintering process at 200 °C, was obtained. It was also noticed that after eight UV/Vis sintering runs, the value of resistivity of silver patterns decreased, and it was as low as four times that of bulk silver [23]. The photothermal sintering process using a low-power UV light source with a wavelength of 395 nm was used to transform the ink composed of silver NPs into solid conductive films with an average resistivity of 0.48 µΩ·m after 30 s of UV irradiation, which was lower in comparison to that after 15 min of oven sintering at 130 °C [24]. Visible light (LED) was also applied to sinter silver nanoparticles for solvent-resistant nanofiltration (SRNF) membrane preparation [25]. A flash-light process was used for sintering the films composed of Ni NPs; however, the obtained resistivity was found to be 76.34 µΩ·cm [26].

In this work, the applicability of the UV-Vis irradiation for the sintering of the coatings composed of nickel–silver core–shell (Ni@Ag) NPs was examined. Ni@Ag NPs with average sizes of 220 nm were deposited on a polyethylene napthalate (PEN) substrate, namely the heat-sensitive substrate, and the UV-Vis sintering process was performed. The effects of the wavelength of the UV-Vis irradiation and the time of sintering on the resistivity of the deposited metallic coatings were investigated. The formation of well-conductive coatings based on Ni@Ag NPs using the UV-Vis sintering process, to the best of our knowledge, is reported for the first time and may represent a new approach for obtaining low-cost printed flexible electronics.

## 2. Materials and Methods

### 2.1. Materials

Nickel sulfate hexahydrate (NiSO_4_·6H_2_O), sodium borohydride (NaBH_4_), sodium carboxymethyl cellulose (NaCMC) with MW 90000, silver nitrate (AgNO_3_), and aminomethyl propanol (AMP) were purchased from Sigma-Aldrich (Poznań, Poland). The wetting agent (BYK 348, silicone surfactant) was produced by BYK-Chemie GmbH (Wesel, Germany).

### 2.2. Fabrication of Metallic Ink

In the process of the preparation of metallic ink, firstly, Ni@Ag NPs were obtained using a synthesis method presented in our previous works [27,28,29]. Firstly, using a “wet” chemical reduction process, nickel NPs were obtained as the core of the core@shell structure. In the next stage, on the surface of synthesized Ni NPs, silver ions were reduced, and as the result of such a transmetalation (displacement) reaction, Ni@Ag NPs were formed. The methodology of the preparation of Ni@Ag NPs was described in detail in our previous papers [27,28,29]. To remove the excess of stabilizer and other additives, the obtained dispersion of Ni@Ag NPs was washed with distilled water and concentrated to 25 wt% via the centrifugation/redispersion process (two times). Then, BYK 348 (0.05 wt%), as a wetting agent, was added, which was followed by an ultrasonication process (30 min at 20 kHz) to obtain homogeneous ink.

### 2.3. Preparation of Conductive Coatings

The ink formulation was deposited on PEN substrates (Kaladex^®^ PEN 2000, DuPont Teijin Films, Redcar, North Yorkshire, UK) with sizes of 0.3 × 0.2 × 0.125 mm by bar coating with the use of K-Hand Coater (Kontech, Łódź, Poland) [30]. Before the ink coating process, the polymeric substrate was washed with isopropanol and distilled water and dried with compressed air. The deposited ink coatings (at the size as the substrate area) were dried on a hot plate at 80 °C for 15 min and finally sintered using a lamp with UV-Vis irradiation (Osram, Warsaw, Poland) at the wavelength regime of 300–700 nm and power of 2.90 mW/cm^2^. The UV-Vis system was equipped with a cooling system to avoid heating the sample during irradiation. The temperature during the UV-Vis sintering process was controlled, and it was 28 °C.

### 2.4. Characterization

The size of Ni@Ag NPs was measured via dynamic light scattering (DLS), using Zetasizer Nano Series (Malvern Instruments, Malvern, Worcestershire, UK) as the value of the three runs with at least 20 measurements at 25 °C. The topographical and morphological properties of the obtained coatings were visualized via an optical microscope (HR-2500) and via Scanning Electron Microscopy (SEM, LEO Gemini 1530, Zeiss, Jena, Germany). The molecular structure and the functional groups of CMC, as well as the components of the coatings based on Ni@Ag NPs before and after the UV-Vis sintering process, were identified using Fourier Transform Infrared Spectroscopy (FTIR) performed using a ThermoFisher Nicolet iN10 microscope cooled with liquid nitrogen at a frequency ranging from 4000 cm^−1^ to 500 cm^−1^. The samples of CMC- and nanoparticle-based films were deposited on the top of the gold-covered glass microscope slides, which were prepared using the Turbo Sputter Coater K575X (Quorum Emitech, South Stour Avenue, Ashford, Kent TN23 7RS). The glass slides were first coated with 10 nm of Cr, followed by 50 nm of Au. The thickness of deposited metallic films was measured via the EDXRF technique (FISCHERSCOPE X-RAY XDL 230, Worcestershire, UK). The values of the sheet resistance of UV-Vis sintered metallic coatings were measured using a four-point probe method (Milliohm Meter, Extech Instruments, Nashua, NH, USA). In this technique, the sheet resistances were automatically measured as four equally spaced measures via the manual contact of the colinear Calvin probes with the coated films, resulting in electrical contact [31]. Then, by multiplying the measured values of the sheet resistance by the thicknesses of the coatings, their resistivities were calculated [32].

## 3. Results and Discussion

In our previous works [27,28] regarding the sintering of metallic coatings composed of Ni@Ag NPs, we mainly focused on thermal sintering; however, a chemical process was also investigated. In the case of thermal sintering (300 °C), the calculated value of the resistivity of films, composed of Ni@Ag NPs at the average size of about 220 nm, was 14.4 µΩ·cm, which corresponds to the conductivity of 48% of that for bulk nickel [28]. In order to avoid such a high-temperature process, the effect of oxalic acid (OA) as a chemical sintering method was investigated [29]. Treating the coatings composed of 220 nm of Ni@Ag NPs with 1% of OA allows the significant decrease in temperature needed to obtain conductive coatings from 300 to 100 °C. Such coatings showed a resistivity of 30 µΩ·cm, only four times higher than that of the bulk nickel. Treatment with OA is a promising approach for the application of such metallic films in flexible electronics. It is also important from an economic point of view, as energy consumption should be as low as possible. In this context, our research seeks to replace the thermal method used for the radiation-induced sintering of metallic coatings. Therefore, in this paper, the effect of the UV-Vis sintering process on the conductive properties of films based on Ni@Ag NPs was studied. The scheme of the fabrication of conductive coatings based on Ni@Ag NPs is shown in Figure 1. As can be seen, such coatings were prepared in a multistep process: ink was formulated, ink was deposited on the polymeric substrate, and, finally, UV-Vis sintering was performed to transform the non-conductive structure into conductive coatings.

The process of the synthesis of Ni@Ag NPs, which involved a two-step reaction, was developed in our laboratory and presented in our previous papers [27,28,29]. In the first step, using a “wet” chemical reduction process, the Ni ions in the complex with AMP and citric acid were reduced, using NaBH_4_ as a reducing agent and in the presence of sodium carboxymethyl cellulose (0.5%) as a stabilizer, to obtain Ni NPs. At the next step, to protect Ni NPs against oxidation, the transmetalation process was performed via the reduction of silver ions (0.04 M) on the surface of obtained Ni NPs, which results in the fabrication of the core–shell structure. According to DLS measurement, the average size (as the value of three subsequent runs of the instrument) of obtained Ni@Ag NPs was found to be 220 nm (Figure 2A). In Figure 2B, the UV-visible spectra of the Ni and Ni@Ag NPs are presented. As can be noticed, the spectrum of Ni@Ag core–shell NPs (dotted line) is characterized by a distinct peak at about 420 nm, typical of the surface plasmons of silver with nano size (in various forms, such as plates, spheres, islands, rings, etc.) [33], while bare Ni NPs have no peaks in the spectrum, which suggest the formation of core–shell structure. In the SEM image (Figure 2C), almost spherical Ni@Ag NPs can be observed. The EDX analysis (Figure 2D) confirmed presence of both nickel and silver. More details about the characterization of such NPs can be found in our recent works [27,28,29].

To obtain uniform and homogenous ink coatings, based on Ni@Ag NPs on PEN substrate, which is an important characteristic for obtaining high conductivity, their properties were optimized via the addition of wetting agents at various concentrations. The morphologies of deposited ink coatings, after the deposition and drying process (80 °C, 15 min), were characterized using an optical microscope (Figure 3). As can be seen in Figure 3A, the structure of the coating without a wetting agent did not show good quality, as it was nonuniform, and some holes can be seen. The most uniform coatings were obtained after addition to ink formulation BYK 348 at a concentration of 0.05%, as is presented in Figure 3B. The structures of such coatings included a lack of holes and cracks, which are the main requirements for film fabrication with high conductivity.

The coatings with optimal morphological properties formed from the ink containing BYK 348 were sintered at various ranges of the wavelength of UV-Vis irradiation. After that step, using the four-probe electrode method, the measurement of the sheet resistance values of the obtained metallic coatings was performed. The obtained values were multiplied by the thicknesses (analyzed using the EDXRF method), which were similar (about 2 µm) for all coated films, and the values of resistivity were calculated. Finally, the conductivities, as the reciprocal values of resistivity, were calculated.

Figure 4A shows the dependence of the resistivity for metallic coatings formed from inks based on Ni@Ag NPs and BYK 348 after the UV-Vis sintering process (after 90 min) in the wavelength range extending from 300 to 700 nm. As can be seen, the values of lowest resistivity (24–25 µΩ·cm) could be obtained for the metallic coatings sintered at the wavelength range 350–400 nm, which corresponds to the surface plasmon resonance (SPR) of silver nanostructures. The excitation of plasmon resonances that generate the localization and enhancement of the electric field at the interface between NPs has been identified as an important effect in the systems [34,35,36]. Electrons emitted from the surface at these enhanced areas give rise to two primary effects: the first effect is to initiate the decomposition of organic stabilizer and the conversion of this material into α-C:H and α-C, while the second effect is to produce a softening of the lattice at the NPs’ surfaces [34]. It is reasonable to expect that all the α-C:H bridges can be replaced by welded necks if the irradiation time is sufficiently long. Therefore, the optimization of the sintering time was performed at a wavelength of 400 nm; the results are presented in Figure 4B.

The values of resistivity decrease from 65 to 22 µΩ·cm upon increasing the UV-Vis sintering time from 15 to 240 min. The optimal value of resistivity, i.e., ~24 µΩ·cm, was obtained after 90 min of irradiation, suggesting that further increasing the time does not show significant changes in resistivity. The calculated conductivity of such optimized film corresponds to 29–31% of that of a bulk nickel (about 7% of that or bulk silver). In the published literature, the research into the fabrication processes of conductive coatings based on Ni@Ag NPs using the UV-Vis sintering process has not so far been presented. However, the most similar ones based on Ni NPs were prepared by Park et al. [26]. Via the optimization of the flash-light sintering parameters, such as light energy and pulsed light patterns, they obtained films with resistivity values of 76.34 µΩ·cm, which were higher than those of the coatings formed in the presented paper (~24 µΩ·cm). The higher conductivity of the coatings obtained in our work could be the result of the presence of the silver shell. The photonic curing of a screen-printed Ni flake ink on solid bleached sulphate paperboard (SBS) and polyethylene terephthalate polymer (PET) substrates was applied for the sensor application [37]. Minimum sheet resistances of 4 Ω/sq on SBS and 16 Ω/sq on PET were obtained. The Ni-based flexible transparent conductive panels with sheet resistances of 53 Ω/sq on various substrates through the LRS of solution-processed NiOx thin films using a laser digital patterning process were obtained by Nam et al. [38]. However, it is difficult to compare such values [37,38] to the resistivity of metallic coatings formed in the presented research because the thicknesses of the obtained films were not measured.

The conductivity of coatings based on Ni@Ag NPs after UV-Vis sintering (at optimal conditions of a time of 90 min and a wavelength of about 400 nm), obtained in the current study, was lower (~30% of that for bulk nickel) in comparison to the value achieved after the thermal sintering process (48% of bulk nickel), which was presented in our previous work [28]. A slightly lower resistivity (35% conductivity of bulk nickel) was also achieved in our recent study of coatings composed of Ni@Ag NPs doped with 1% of Ag NPs but at a sintering temperature of 150 °C [27]. However, the applied temperature in the mentioned works [27,28] was limiting for heat-sensitive substrates. Moreover, using the sintering process without heating is more profitable from an economic point of view. In the current research, the conductivity of sintered coatings was higher than that after using oxalic acid as a chemical agent (23% conductivity of bulk nickel) [29]. Therefore, the UV-Vis sintering method could be more promising with regard to the application of obtained conductive materials in the fabrication of flexible circuits and devices.

To define the possible mechanism of the UV-Vis sintering process for coatings based on Ni@Ag NPs, the FTIR measurement before and after the irradiation of formed coatings was performed. Moreover, to better understand the system influence of UV-Vis irradiation, the polymeric film formed with pure CMC was investigated

The IR spectra of CMC (0.5%, concentration used in synthesis of Ni NPs) films before and after heating at 80 °C (15 min) and irradiation via UV (90 min) are shown in Figure 5, in which the differences in intensity of both spectra are visible. The strong and broad absorption band characteristics of CMC can be observed in the vibration bands at around 3412 cm^−1^ and 2923 cm^−1^, which are assigned to the stretching vibrations of the –OH and C-H groups, respectively. The peak at 1604 cm^−1^ confirms the presence of carboxyl groups (COO–), and the bands at around 1419 cm^−1^ and 1325 cm^−1^ are assigned to CH_2_ scissoring and –OH bending vibration, respectively. The stretching vibration band at 1064 cm^−1^ comes from –CH-O-CH_2_ group [39,40]. For CMC treated with temperature and UV light, the intensity of infrared spectra decreased, and a shift in the maximum band vibration of the hydroxyl group from 3412 cm^−1^ to 3393 cm^−1^ was observed. Also, the stretching vibration of C-H from 2923 cm^−1^ for pure CMC shifts to 2915 cm^−1^ for CMC heated and irradiated via UV irradiation was observed. According to Wang et al. [39], this phenomenon can occur due to the decrease in the crystallinity index of CMC caused by the breaking of some hydrogen bonds in the crystalline parts of CMC chains. Thus, the sintering process of coatings based on Ni@Ag NPs can be attributed to their destabilization due to the lowering of the crystallinity index of CMC.

In Figure 6, the FTIR spectra of CMC and Ni@Ag NPs coatings after drying at room temperature and film based on Ni@Ag NPs after drying at 80 °C (15 min) and UV-Vis sintering (90 min) are compared. As can be seen, the FTIR spectra of coatings, before and after UV-Vis irradiation, formed from Ni@Ag NPs stabilized with CMC (Ni@Ag-CMC) show big decreases in absorption band intensity in the whole measured range in comparison to that obtained for pure CMC film dried at room temperature. The shift and band intensity changes in the range of 3300–3500 cm^−1^ indicate the possible means of inorganic–organic material connection (Ni@Ag-CMC) [39], which can explain the stabilization of nanoparticles by CMC. The intensity of the characteristic absorption bands for CMC at 1604 cm^−1^ and 1419 cm^−1^, assigned to the asymmetric and symmetric stretching vibrations of carboxyl groups (COO–), decreased to 1587 cm^−1^ and 1416 cm^−1^, respectively, for Ni@Ag NP-based coatings. It can be considered to be a characteristic of carboxylic acids adsorbed on NPs [41]. Moreover, for the spectrum of Ni@Ag-CMC treated with UV light, the peak intensity of the band near 1262 cm^−1^ and 1064 cm^−1^ can hardly be observed compared to that of CMC owing to ether bond cleavage after the introduction of CMC to the NP structure. Thus, more highly anionic CMC can interact more strongly with free Ag ions (if they exist in Ni@Ag NP-based coatings) through not only negatively charged COO_2_ groups but also ionized OH_2_ groups. The large number of these anions in CMC can facilitate the reduction of Ag ions to Ag NPs when exposed to UV light and improve the UV-Vis sintering process of coatings based on Ni@Ag NPs [42].

The changes in the intensity of absorption bands for the coatings composed of Ni@Ag NPs before and after irradiation with UV light are less visible. For both types of analysis (before and after the UV-Vis sintering process), the intensity of the main hydroxyl group at 3393 cm^−1^ almost vanished, and its shift to 3443 cm^−1^ can be seen. This can be attributed to the decreasing crystallinity of CMC [39]. However, even if there are almost no differences in the absorption bands in the measurement range of 3000–2000 cm^−1^ in the spectra of coatings based on nanoparticles before and after UV-Vis sintering, those obtained at room temperature were not conductive. It is also worth mentioning that all FTIR measurements were performed for coatings; therefore, the analysis and interpretation of the results are more complicated than for the solutions. Therefore, it is difficult to clearly indicate the mechanism of the UV-Vis sintering process of films composed of Ni@Ag NPs.

Another suggested mechanism for the sintering of Ni@Ag NP-based coatings is based on Prajapat et al. [43] work, and these authors noticed that during UV irradiation, generated hydroxyl radicals attack the long polymer chains of CMC and result in viscosity reduction. UV irradiation can also help to initiate the breakage of the backbone chain of the polymer, which results in the lowering of the molecular weight of the polymer and its depolymerization without big changes in FTIR spectra after this process. This phenomenon can also lead to the destabilization of Ni@Ag NPs in sintered coatings and, thus, be responsible for their conductivity.

The morphology and topography of metallic coatings obtained after the drying process (80 °C, 15 min), as well as after UV-Vis irradiation treatment at optimal conditions (time of 90 min and wavelength of 400 nm), were analyzed via the SEM technique. As can be seen in Figure 7, the NPs after the UV-Vis sintering were much more tightly connected or even welded compared to a coating that was only dried, resulting in an interconnected network. The mechanism of this sintering has not been reported in the literature and needs further study. However, it is clear from the above observations that UV-Vis irradiation may be a useful technique for transforming a non-conductive coating based on metallic NPs into a conductive coating. More information concerning the coalescence between NPs, e.g., how the neck layers form, can be found in the literature [34].

## 4. Conclusions

The results presented in this paper suggest that UV-Vis irradiation is an effective method of sintering Ni@Ag NPs, being suitable for obtaining conductive metallic coatings from NP-based inks. The calculated conductivity of such coatings corresponds to 29–30% of that of a bulk nickel. To the best of our knowledge, this is the first time that coatings composed of Ni@Ag NPs have been sintered via UV-Vis irradiation and such low resistivity/high conductivity has been obtained. In contrast to thermal sintering, UV-Vis irradiation is a promising method for the preparation of electronic tracks on heat-sensitive substrates, like papers and plastics; however, more detailed studies of its mechanism are still required.

## Figures and Tables

**Figure 1 materials-16-07218-f001:**
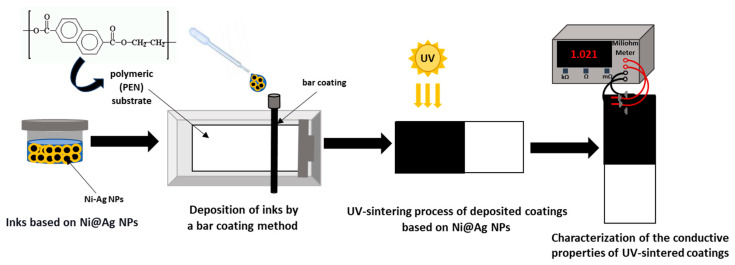
Scheme of the fabrication of the conductive coatings based on Ni@Ag NPs by using UV-Vis sintering process.

**Figure 2 materials-16-07218-f002:**
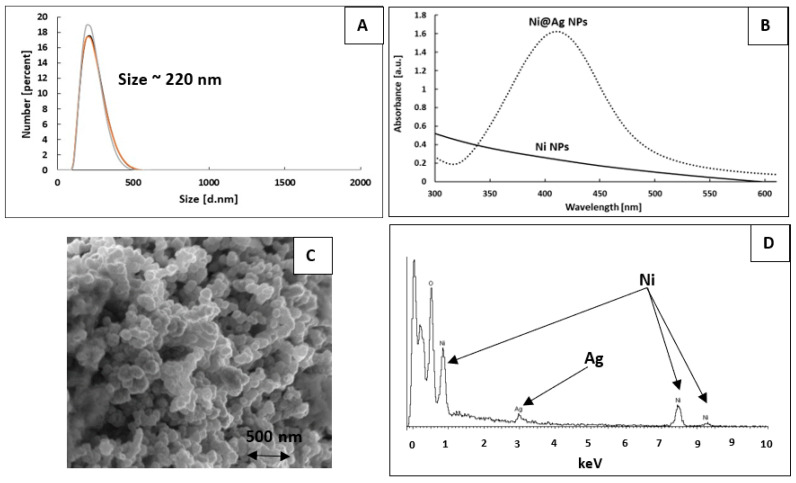
The characterization of the dispersion of the Ni@Ag NPs: size distribution (**A**), UV-Vis spectrum ((**B**), dotted line), SEM image (**C**), and EDX spectrum (**D**). The UV-Vis spectrum of the dispersion of the Ni NPs ((**B**), solid line).

**Figure 3 materials-16-07218-f003:**
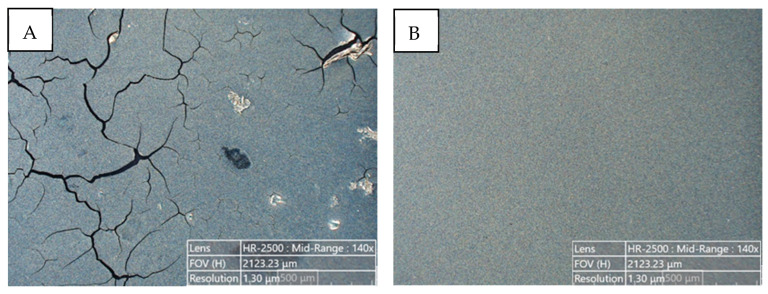
Examples of optical microscopy images of deposited ink coatings based on Ni@Ag NPs after the drying process (80 °C, 15 min): (**A**) without a wetting agent; (**B**) with BYK 348 at 0.05%.

**Figure 4 materials-16-07218-f004:**
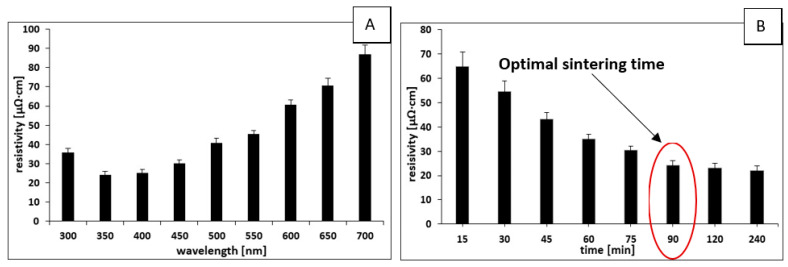
The dependence of the resistivity of coatings formed from Ni@Ag ink after the UV-Vis sintering process (**A**) at a time of 90 min in the wavelength range 300–700 nm and (**B**) at a wavelength of 400 nm at various sintering times.

**Figure 5 materials-16-07218-f005:**
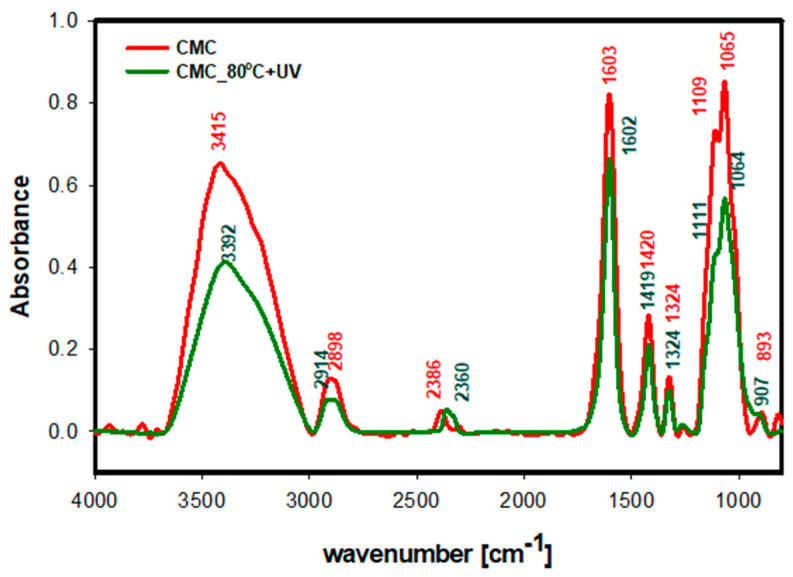
FTIR spectra of CMC films: dried at room temperature (red); dried at 80 °C and irradiated with UV for 90 min (green).

**Figure 6 materials-16-07218-f006:**
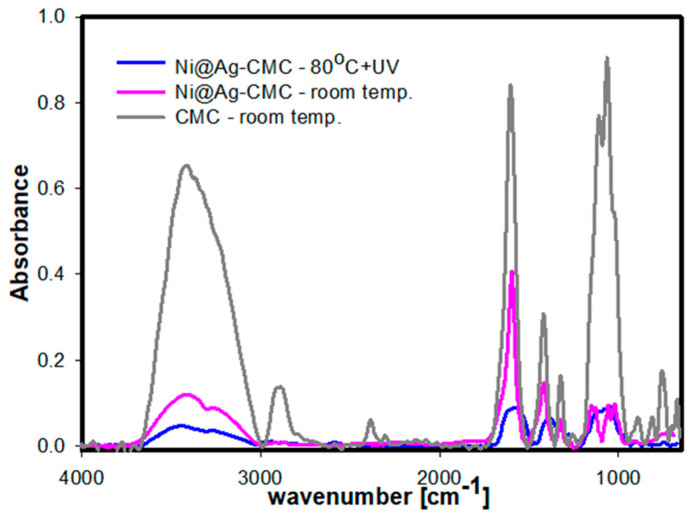
The comparison between the FTIR spectra of CMC (grey) and Ni@Ag NP (pink) coatings after drying at room temperature and films based on Ni@Ag NPs after drying at 80 °C (15 min) and UV-Vis sintering for 90 min (blue).

**Figure 7 materials-16-07218-f007:**
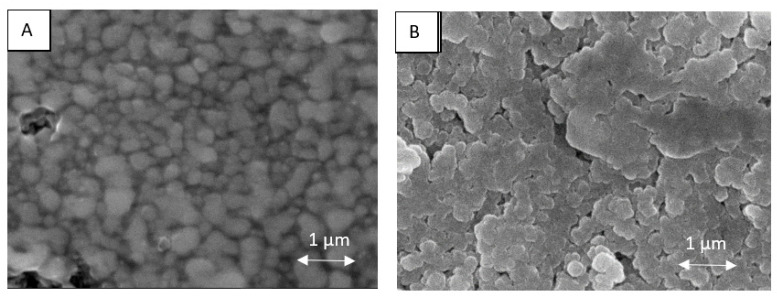
Examples of SEM images obtained for coatings based on Ni@Ag NPs after drying (**A**) and the UV-Vis sintering process at the optimal time conditions time—90 min and wavelength of 400 nm (**B**).

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
