# Peer review of "UV-Vis Sintering Process for Fabrication of Conductive Coatings Based on Ni-Ag Core–Shell Nanoparticles"

_materials, 2023, doi:10.3390/ma16227218_

Round 1

Reviewer 1 Report

Comments and Suggestions for Authors

The manuscript entitled "UV-Vis Sintered Conductive Coatings based on Ni-Ag Core-2 Shell Nanoparticles for Flexible Electronics Application" is focused on the problem of fabrication of conductive nickel layers with conductivity close to metallic Ni by UV-Vis sintering of precursors based on Ni-Ag core-shell nanoparticles with additives to optimise this process.

 This study is of interest to the readers of the journal Materials, but before the manuscript can be considered further, it is necessary to comment and answer some important issues. 

1. In the Introduction, the authors refer to the methods where laser irradiation is applied. It is recommended to include references to actual laser techniques (10.3390/ma14102493 ; 10.3390/nano13192693 ; 10.1039/D1TC00435B) and to make a comparison with the one proposed by the authors, especially regarding the speed of the process, the selectivity and the variability of the substrates.  For example, does the process described depends on the type of substrate. 

2. It is important to point out, in the UV sintering process, does the deposit cover the full substrate or only a defined area, which has been previously prepared? 

3. The quality of the IR spectra in Figs. 5 and 6 can be enhanced.

4. Did you estimate which temperatures are reached during the process of formation of the conductive layer? 

This question is important to understand the mechanism. It is also important to point out how the process in this study compares to temperature sintering in other paper by the authors. To emphasise how the authors' present work differs from previous studies cited in the reference list. 

5. Since authors consider SPR to be the key initiator of the sintering process, it would be useful to illustrate if there is a dependence on the amount of silver nanoparticles added.  

6. To improve the investigation, the authors are invited to provide a proof-of-concept of the method for prototyping the electronic devices reported in the Title. 

Reviewer 2 Report

Comments and Suggestions for Authors

The UV-Vis sintering process was applied for the fabrication of conductive coatings composed of low-cost nickel-silver nanoparticles (NPs) with core-shell structures. The metallic films were formed on a plastic substrate (polyethylene naphthalate. Conductivity corresponding to about 30% of that for bulk nickel was obtained for the coatings. The paper could be accepted after revision.

-Ni@Ag NPs were obtained by using a synthesis method presented in previous works of the authors. So, it is not useful to describe the procedure here again.

- Structure of polyethylene napthalate should be demonstrated.

- Figure 7 B is not in the figure ?

- The calculated conductivity of such coatings corresponds to 29-30% of that for a bulk nickel. Why the coductivity is compared with that of bulk nickel ? The described coatings have nickel-silver particles ?

- The conductivity should be compared with other similar conductive coatings, which are described in literature. Advantages and disadvantages should be described and compared.

-Application of the coating in practical products could be demonstrated ?

Reviewer 3 Report

Comments and Suggestions for Authors

In this work, Anna Pajor-Åšwierzy et al. examined the applicability of the UV-Vis irradiation for the sintering of the coatings composed of nickel-silver core-shell (Ni@Ag) NPs. Ni@Ag NPs with an average size of 220 nm were deposited on polyethylene napthalate (PEN) substrate, the heat-sensitive substrate, and the UV-Vis sintering process was performed. The effects of the wavelength of the UV-Vis irradiation and the time of sintering on the resistivity of the deposited metallic coatings were investigated. However, the flexible electronics application is missing. It is difficult to match with the title information. In addition, other issues should be solved before further consideration, as below:

1) The novelty of this work is not clear;

2) The authors previously reported similar work (Colloids Interfaces 2021, 5, 15; Materials 2021, 14, 2304; Materials 2022, 15, 305). The difference betwen this current work and the previous work should be emphasized. 

3) The Introduction is fuzzy. Please rephase them. 

4) Fig. 2B, please provide the EDX information of the as-prepared film. The SEM image is not enough. Fig. 2C just presents its UV-visible spectra. 

5) Fig. 2A and 2C, please use the data processing tool to present data with a uniform format. This comment is not required, but recommonded. 

6) Fig. 3, the caption and scale is not clear. 

7) Fig. 4, the same suggestion with Question 5.

8) Line 200-205, this statement is not correct. Figure 2C is  UV-visible spectra. How can obtain the resistivity info? The key equation is necessary. 

9) Fig.5 and Fig. 6 are fuzzy. Please improve the image resolution. 

10) Fig. 7, the image is not complete. The caption included A and B, but missing. 

More important, no flexible electronics application in the whole manuscript. if insist, please change the title in the manuscript. 

Round 2

Reviewer 2 Report

Comments and Suggestions for Authors

 Accept in present form

Reviewer 3 Report

Comments and Suggestions for Authors

The revised manuscript is good, it can be accepted by Materials, if other reviewers have no negative comments.